# How to Predict Outcomes from a Biofeedback and Pelvic Floor Muscle Electric Stimulation Program in Patients with Urinary Incontinence after Radical Prostatectomy

**DOI:** 10.3390/jcm11010127

**Published:** 2021-12-27

**Authors:** Stefano Salciccia, Alessandro Sciarra, Martina Moriconi, Martina Maggi, Pietro Viscuso, Davide Rosati, Marco Frisenda, Giovanni Battista Di Pierro, Vittorio Canale, Giulio Bevilacqua, Gianluca Nesi, Francesco Del Giudice, Alessandro Gentilucci, Susanna Cattarino, Gianna Mariotti

**Affiliations:** 1Department of Maternal-Infant and Urologic Sciences, Sapienza Rome University, Policlinico Umberto I Hospital, Viale del Policlinico 151, 00161 Rome, Italy; stefano.salciccia@uniroma1.it (S.S.); martina.moriconi@uniroma1.it (M.M.); sciarrajr@hotmail.com (M.M.); pietro.viscuso@uniroma1.it (P.V.); davide.rosati@uniroma1.it (D.R.); marco.frisenda@uniroma1.it (M.F.); giovannibattista.dipierro@uniroma1.it (G.B.D.P.); vittorio.canale@uniroma1.it (V.C.); giuliobevilacqua@hotmail.it (G.B.); francesco.delgiudice@uniroma1.it (F.D.G.); alegenti@yahoo.it (A.G.); susycat84@hotmail.it (S.C.); mariotti.gianna@gmaill.com (G.M.); 2Department of Urology, Sant’Andrea Hospital, 00141 Rome, Italy; gianlucanesi@hotmail.com

**Keywords:** biofeedback, electric stimulation, pad test, pelvic floor muscle training, radical prostatectomy, urinary incontinence

## Abstract

Objectives: The objective of this study was to analyze the pre-operative and intra-operative variables that can condition urinary incontinence (UI) after radical prostatectomy (RP), as well as continence rate recovery during a pelvic floor rehabilitation program. Materials and Methods: A total of 72 cases with UI after RP were prospectively examined. All cases were homogeneously treated by the same surgeon, using the same RP technique. A combination of biofeedback (BF) and pelvic floor electric stimulation (PFES) performed by the same clinician and using the same protocol was used. Clinical, pathologic and surgical variables were analyzed in terms of 24 h pad test results (pad weight and pad-free status). Results: Prostate volume (PV) strongly varied from 24 to 127 cc (mean ± SD 46.39 ± 18.65 cc), and the baseline pad weight varied from 10 to 1500 cc (mean ± SD 354.29 ± 404.15 cc). PV strongly and positively correlated with the baseline pad weight (r = 0.4215; *p* = 0.0269) and inversely with the three-month pad weight (r = − 0.4763; *p* = 0.0213) and pad-free status (r =− 0.3010; *p* = 0.0429). The risk of a residual pad weight >10 g after the rehabilitative program significantly increased according to PV (*p* = 0.001) and the baseline pad weight (*p* = 0.002 and < 0.0001). In particular, PV > 40 cc and a baseline pad weight >400 g significantly (*p* = 0.010 and *p* < 0.0001, respectively) and independently predicted a 5.7 and a 35.4 times increase in the risk of a residual pad weight at the three-month follow-up, respectively. Conclusion: This is the first prospective trial whose primary objective is to verify the possible predictors, such as PV, that are able to condition the response to a pelvic floor rehabilitation program for UI after RP.

## 1. Introduction

Although there have been improvements in radical prostatectomy (RP) for prostate cancer (PC), this surgical procedure remains significantly associated with the development of urinary incontinence (UI). The rates of UI after RP significantly vary from 5 to 40% in different trials, depending on the characteristics of the populations and on the methods used [1,2,3]. UI is a relevant side effect after RP, and it can develop early after catheter removal and influence the quality of life of men. Non-invasive therapies are often prescribed first, and pelvic floor muscle exercises (PFMEs) can be used to improve the strength of the pelvic floor [1]. The European Association of Urology (EAU) guidelines [1] underline that PFME may speed up the recovery of continence after surgery, and a specific biofeedback (BF)-guided program [4], a pelvic floor electrical stimulation (PFES) [1,2,5,6,7] or their combination have been proposed. A recent meta-analysis showed that guided BF and/or PFES improves early continence recovery when compared to PFME alone [8]. Several pre-operative and intra-operative variables may condition UI after RP, such as continence recovery results after treatments [9]. However, most of the clinical trials published in the literature do not consider these variables [4,5,10,11,12,13,14,15,16,17,18,19,20,21,22,23,24,25,26,27,28,29,30], such as preoperative conditions, co-morbidities, prostate volume and the surgical techniques of RP. The heterogeneity of UI levels (pad weight), detected in a meta-analysis of baselines after RP, is likely conditioned by some of these variables [8].

## 2. Materials and Methods

### 2.1. Objectives

The *aim* of the present prospective trial was to analyze the pre-operative and intra-operative variables that can condition UI after catheter removal at RP and continence rate recovery during a pelvic floor rehabilitation program. In particular, in a population of PC cases with UI after RP, we prospectively evaluated whether pre-operative clinical characteristics, surgical techniques of RP and pathologic outcomes are able to significantly and independently influence baseline pad weight, pad weight improvement and pad-free status during the treatment with a combined program of BF and PFES and at 3-month follow-up.

### 2.2. Patients

This is a prospective trial on PC patients submitted to RP with post-operative UI. A real-life setting was analyzed at our Urological Department, using homogeneous criteria for the management of PC and UI.

Patients with a histological diagnosis of prostatic adenocarcinoma considered for RP as a primary therapeutic option and presenting persistent UI at 30 days after catheter removal were consecutively included in the analysis. The protocol was approved by our internal ethical committee, and all patients gave their informed consensus for each procedure. All diagnostic and therapeutic procedures reflected our routine clinical practice in a department at high volume for the management of PC disease. Inclusion criteria were the following: histological diagnosis of adenocarcinoma, no distant metastases at clinical staging, RP chosen as primary treatment option, estimated life expectancy of ≥10 years, persistent UI (urinary leakage > 5 g at pad test) at 30 days after catheter removal and referred by the patient as able to influence his quality of life. Exclusion criteria were the following: prior bladder or prostate surgery, prior urinary of fecal incontinence, neurogenic dysfunctions, history of overactive bladder, neurological conditions, psychiatric therapies or other drugs able to influence bladder function, peri-operative complications, post-operative urinary stricture and early post-operative PC progression with the need for adjuvant treatments. No patient was prescribed anticholinergic drugs (or other drugs able to influence urinary continence) during the period of analysis. From January 2019 to January 2021, seventy-two consecutive cases corresponding to our inclusion and exclusion criteria were included in the study.

### 2.3. Clinical and Pathologic Parameters

The whole population of 72 cases is described in Table 1. Clinical characteristics of our population, including comorbidities, were obtained in all cases. Before surgery, clinical staging and risk category (D’Amico and EAU classification) assessment were performed using total prostate-specific antigen (PSA) determination and imaging (multiparametric magnetic resonance (mpMRI), CT and bone scan) [1]. Prostate volume (PV) was homogeneously determined at mpMRI and confirmed on the pathologic specimen using the ellipsoid estimation. All patients were submitted either to a laparoscopic approach (LRP) or a robotic RARP approach, following EAU guidelines for indications [1]. As routine clinical practice in our departments, each procedure (RARP and LRP) was discussed with the patient and performed by the same surgeon who had a high expertise (more than 10 years of experience and 500 procedures performed) in each approach, consistent with best practice. All surgical procedures were performed using the same intraperitoneal standard technique for RP. A nerve-sparing (NS) (intrafascial, monolateral or bilateral) procedure was performed based on risk classes and the risk of extracapsular disease [1]. In particular, for both RARP and LRP, patients with clinical high risk of ipsilateral extracapsular disease were excluded from NS surgery; extended lymph node dissection (eLND) was performed in all high-risk cases and in the intermediate-risk class in cases with ≥5% probability for positive nodes [1]. After surgery, the catheter was homogeneously removed in all cases at an interval between 7 and 10 days. Gleason score and grade groups according to the World Health organization (WHO)/ISUP 2014 guidelines at surgery, pathologic staging using TNM classification and surgical margin (SM) status were routinely defined in all cases.

### 2.4. Conservative Treatment for Urinary Incontinence

All cases were submitted to the same program of BF + PFES performed by the same clinician with more than 10 years of experience. In all cases, the program started 30 days after catheter removal and after the determination of baseline pad weight. The BF + PFES program has been described in previous articles [18,24]. Patients met the clinician twice a week, and treatment sessions were homogeneously composed of a first part with BF (15 min) followed by a second part with PFES (20 min) for a total of 35 min. For PFES, a surface electrode was inserted into the anus and pulsed at 30 Hz (first 10 min) and 50 Hz (second 10 min) square waves at a 300 μs pulse duration and a maximal output current of 24 mA. The intensity was adequate to induce visual lifting of the levator ani and pubococcygeus muscles, considering the level of comfort of the patient. For BF, a 2-channel electromyographic BF apparatus was used, with 1 channel for perineal and the other for abdominal muscle, and the signal was received through surface electrodes.

### 2.5. Parameters in Terms of Continence Recovery

Thirty days after catheter removal (baseline), during the treatment (BF + PFES) and at 3-month follow-up, urinary continence status was homogeneously assessed in terms of the 24 h pad test and pad weight (in grams). No pad use (pad-free status) or less than 2 g at pad test defined continence [1].

### 2.6. Statistical Analysis

For statistical evaluation, the SPSS Statistics 1.6 program (StataCorp, College Station TX, USA) was used. Descriptive statistical methods, such as number of cases, mean ± SD, median and range, were used. For the comparison of quantitative data and pairwise intergroup comparisons of variables, a Mann–Whitney test, Student’s t-test and one-way ANOVA test were performed. Pearson correlation analysis was also performed. Univariate and multivariate Cox proportional analyses to show the significant and independent role of the different variables in determining continence recovery were used. Primary outcomes were the following: baseline pad weight at catheter removal after RP, pad weight variation during pelvic floor program and pad-free status at 3-month follow-up. Statistical significance was evaluated at *p* < 0.05.

## 3. Results

### 3.1. Baseline Characteristics of the Population

The baseline characteristics of our population are described in Table 1. In particular, age ranged from 49 to 75 years (mean ± SD 65.73 ± 4.93), 70.8% of cases presented a mild metabolic syndrome, and PV strongly varied from 24 to 127 cc (mean ± SD 46.39 ± 18.65). Sixty-four (88.9%) cases underwent LRP and eight (11.1%) RARP procedures; an NS technique was performed in fifteen (20.8%) cases. At the final pathologic evaluation, an extracapsular disease (pT3) was found in 26.4% of cases, and positive SM was found in nine (12.5%) cases. At the 30-day follow-up after catheter removal, baseline pad weight strongly varied from 10 to 1500 cc (mean ± SD 354.29 ± 404.15 cc). The number of sessions of PF rehabilitation was in the range of 6–22 (mean ± SD 12.40 ± 4.87).

### 3.2. Correlation Analysis

According to Pearson analysis (Appendix A), only metabolic syndrome and PV significantly correlated with baseline pad weight and pad test results after rehabilitation. In particular, the presence of a metabolic syndrome positively correlated with baseline pad weight (r = 0.2639; *p* = 0.0466) and inversely with pad-free status at follow-up (r = −0.2433; *p* = 0.0418). PV strongly and positively correlated with baseline pad weight (r = 0.4215; *p* = 0.0269) and inversely with 12-week pad weight (r = −0.4763; *p* = 0.0213) and pad-free status (r = −0.3010; *p* = 0.0429).

### 3.3. Analysis in Terms of Prostate Volume Stratification

The whole population was stratified on the basis of PV into two groups: ≤40 cc (40.3% of cases) and >40 cc (59.7% of cases) (Table 2). According to this stratification, baseline pad weight significantly (*p* = 0.0009) varied between the two groups, with a median value of 174 cc in cases with PV ≤ 40 cc and of 360 cc in cases with PV > 40 cc. The pad weight during rehabilitation treatment and follow-up was always significantly (*p* < 0.01) lower in the group with PV ≤ 40 cc, with a higher percentage of reduction from baseline to 12-week follow-up (PV ≤ 40 cc = 94.3%; PV > 40 cc = 85.1%) and, finally, a higher percentage of pad-free cases (PV ≤ 40 cc = 55.2%; PV > 40 cc = 27.9%) (Figure 1a,b).

### 3.4. Analysis in Terms of Baseline Pad Weight Stratification

The population was stratified according to the baseline pad weight after 30 days from catheter removal into three groups: <100 g (29.2% of cases), 101–400 g (34.7% of cases) and >400 g (36.1% of cases) (Table 3). No significant (*p* > 0.05) difference in terms of clinical and pathologic characteristics among the three groups was found, except for PV, which progressively and significantly (*p* = 0.0001) increased according to the baseline pad weight. A significantly (*p* = 0.0499) lower percentage of cases submitted to the NS RP technique showed an elevated baseline pad weight over 400 g (3.8%) when compared to cases with a baseline pad weight <100 g (28.6%). Pad weight during rehabilitation treatment and follow-up was always significantly (*p* < 0.0001) lower in the group with a baseline pad weight < 100 g, with a higher percentage of reduction from baseline to 12-week follow-up (baseline pad weight < 100 g = 92.6%; baseline pad weight 101–400 g = 85.0% and baseline pad weight > 400 cc = 86.5%) and, finally, a higher percentage of pad-free cases (baseline pad weight < 100 g = 76.2%; baseline pad weight 101–400 g = 4.0%; baseline pad weight > 400 g = 8.0%) (Figure 1b).

### 3.5. Analysis in Terms of PAD-free Results

A higher percentage of cases with PV < 40 cc (55.2%) and a baseline pad weight < 100 g (76.2%) reached a pad-free status at the 3-month follow-up. No other clinical or pathologic variables were able to significantly (*p* > 0.05) influence pad-free outcome after the rehabilitation program. Pad weight was always significantly (*p* < 0.001) lower at each follow-up interval in cases where pad-free status was obtained, and the percentage of pad weight reduction from baseline to 3-month follow-up was 99.8% (Appendix A and Appendix A).

### 3.6. Univariate and Multivariate Analyses

Table 4 shows the logistic regression analysis utilized to identify the variables that are able to predict and to condition a residual pad weight > 10 g at the end of follow-up (three-month follow-up). In the univariate analysis, the risk of a residual pad weight > 10 g did not significantly vary according to most of the clinical, pathologic and surgical variables, whereas it significantly increased according to PV (*p* = 0.001) and the baseline pad weight (*p* = 0.002 and <0.0001). In particular, in the multivariate analysis, PV > 40 cc and a baseline pad weight > 400 g significantly (*p* = 0.010 and *p* < 0.0001, respectively) and independently predicted a 5.7 times and a 35.4 times increase in the risk of a residual pad weight > 10 g at the three-month follow-up, respectively.

## 4. Discussion

To the best of our knowledge, this is the first prospective clinical trial whose primary objective was to verify the possible predictors able to condition the response to a pelvic floor rehabilitation program for UI after RP. EAU guidelines underlined that there is some conflicting evidence on whether the addition of BF increases the effectiveness of PFME alone and whether PFES may add benefit in the short term [1]. In a recent meta-analysis [8], the use of guided programs (BF and/or PFES) demonstrated a significant positive effect on early continence recovery following RP compared to PFME alone. However, the authors described a significant heterogeneity of results in terms of pad weight (I2 > 80%) among different rehabilitative treatments. Several pre-operative and intra-operative variables may condition UI after surgery. In most studies [10,11,12,13,14,15,16,17,18,19,20,21,22,23,24,25,26,27,28,29,30], data regarding pre-operative conditions, co-morbidities, PV and surgical techniques were not considered or were incompletely described. It is possible that some of these variables conditioned the heterogeneity of UI levels (pad weight) after catheter removal. In the different studies [10,11,12,13,14,15,16,17,18,19,20,21,22,23,24,25,26,27,28,29,30] included in this meta-analysis [8], after surgery, the mean pad weight extremely varied from 7.0 ± 56.3 to 738.5 ± 380.6 g. This is a relevant point because the baseline pad weight amount is a variable that is able to condition results in terms of pad weight improvement at different follow-ups after a PF rehabilitative program.

The strengths of this study are the following: (I) the present trial prospectively considered, as the primary objective, the analysis of predictors and variables that are able to condition results after a rehabilitative program for UI after RP; (II) all cases were homogeneously treated by the same surgeon with a high level of experience, using the same surgical technique for RP; (III) all cases underwent catheter removal after RP and started a PF rehabilitative program at the same time interval; (IV) all programs were homogeneously performed by the same clinician with a high level of experience, using the same protocol for BF + PFES; (V) the use of the 24 h pad test with pad weight determination, which is an objective and recommended test.

However, some limitations deserve mention: (I) the number of cases could be higher and (II) the follow-up could be prolonged. However, most of the changes during these rehabilitative programs are obtained within the first 3 months.

In our experience, most of the clinical, pathological and surgical variables are not able to significantly condition the results of a BF + PFES program for UI after RP. PV is the main variable significantly correlated with pad weight results, and an increased PV is able to significantly and independently increase the risk of higher baseline post-operative pad weights, as well as the risk of residual pad weight > 10 g after the rehabilitative program. A significantly (*p* < 0.01) higher percentage of cases with PV < 40 cc (55.2%) and a baseline pad weight < 100 g (76.2%) reaches pad-free status at the three-month follow-up. 

Future studies should always stratify results in terms of pre-operative variables, in particular, prostate volume and post-surgical pad weight, in order to better understand the results among the different non-invasive treatment strategies.

## Figures and Tables

**Figure 1 jcm-11-00127-f001:**
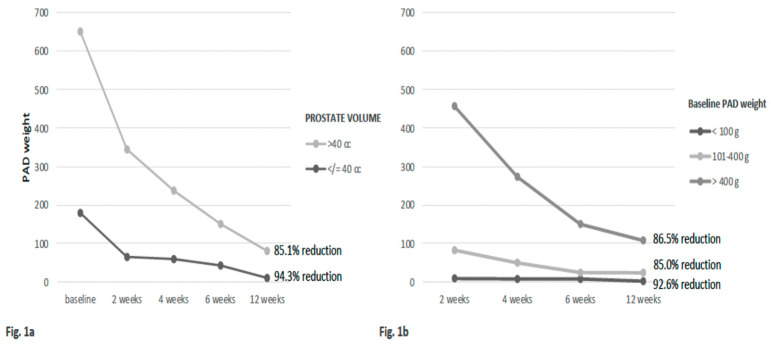
Pad weight variation (mean values) from baseline to 12-week follow-up according to (**a**) prostate volume (PV) and (**b**) baseline pad weight stratification. One-way ANOVA (*p* < 0.01); measurements (338 × 190 mm (150 × 150 DPI)).

**Table 1 jcm-11-00127-t001:** Characteristics of the population submitted to pelvic floor rehabilitation for urinary incontinence after radical prostatectomy.

Patients, *no.*	72
Age (years)	
mean ± SD	65.7 ± 4.9
median (range)	67 (49–77)
Weight (kg)	
mean ± SD	81.4 ± 9.2
median (range)	82.5 (62–108)
BMI	
mean ± SD	25.6 ± 2.3
median (range)	25 (21.0–35.0)
Metabolic Syndrome, no. (%)	
no	11 (15.3)
mild	51 (70.8)
full	10 (13.9)
Prostate Volume (cc)	
mean ± SD	46.4 ± 18.5
median (range)	45 (24–127)
Presence of intravesical prostatic lobe, no. (%)	16 (22.2)
Pre-operative total PSA (ng/mL)	
mean ± SD	8.0 ± 3.8
median (range)	7.4 (2.0–23.0)
Post-operative total PSA (ng/mL)	
mean ± SD	0.05 ± 0.13
median (range)	0.03 (0.01–0.8)
NS technique at RP, no. (%)	15 (20.8)
eLND performed at RP, no. (%)	14 (19.4)
Pathological stage (T), no. (%)	
pT2	53 (73.6)
pT3 a	15 (20.8)
pT3 b	4 (5.6)
Surgical technique at RP, no. (%)	
- LRP	64 (88.9)
- RARP	8 (11.1)
Positive SM at surgery (R1), no. (%)	9 (12.5)
ISUP grading, no. (%)	
1	19 (26.4)
2	31 (43.1)
3	14 (19.4)
4	7 (9.7)
5	1 (1.4)
Rehabilitation: number of procedures	
mean ± SD	12.4 ± 4.9
median (range)	12 (6–22)
Rehabilitation: time length (weeks)	
mean ± SD	6.3 ± 2.4
median (range)	6 (3–11)
Pad weight at baseline (g)	
mean ± SD	354.3 ±404.1
median (range)	170 (10–1500)
Pad weight at 2 weeks (g)	
mean ± SD	192.3 ± 250.6
median (range)	70 (0–1029)
Pad weight at 4 weeks (g)	
mean ± SD	136.1 ± 181.4
median (range)	48.5 (0–757)
Pad weight at 6 weeks (g)	
mean ± SD	89.8 ±116.3
median (range)	43 (0–408)
Pad weight at 12 weeks (g)	
mean ± SD	46.2 ± 84.7
median (range)	8 (0–420)
Pad-free cases at 12 weeks, no. (%)	28 (38.9)

RP = radical prostatectomy; BMI = body mass index; PSA = prostate-specific antigen; LRP = laparoscopic RP; RARP = robot-assisted RP; eLND = extended lymph node dissection; ISUP = International Society of Urological Pathology; SM = surgical margin; NS = nerve sparing. (mean ± SD, median (range). Number of cases (%).

**Table 2 jcm-11-00127-t002:** Comparison of PAD weight results on the basis of prostate volume (Two-tailed *t*-test).

	Prostate Volume		*p* Value
	≤40 cc	40 cc	
Patients, no. (%)	29 (40.3)	43 (59.7)	-
Rehabilitation: number of procedures mean ± SD median	9.8 ± 4.1 9	14.1 ± 4.6 15	<0.0001
Rehabilitation: time length (weeks) mean ± SD median	5.1 ± 2.0 5	7.2 ± 2.3 8	<0.0001
Pad weight at baseline (g) mean ± SD median	179.4 ± 262.0 174	472.2 ± 441.6 360	0.0009
Pad weight at 2 weeks (g) mean ± SD median	64.8 ± 103.7 20	280.4 ± 283.6 165.5	<0.0001
Pad weight at 4 weeks (g) mean ± SD median	59.4 ± 94.4 25	178.7 ± 204.1 85.5	0.0017
Pad weight at 6 weeks (g) mean ± SD median	42.4 ± 59.9 16	108.1 ± 127.9 45	0.0083
Pad weight at 12 weeks (g) mean ± SD median	10.2 ± 24.8 0	70.5 ± 101.1 24	<0.0001
Pad weight percentage reduction from baseline to 12 weeks, *%*	94.3%	85.1%	
Pad-free cases at 12 weeks, *no. (%)*	16 (55.2)	12 (27.9)	0.0041

Mean ± SD, median; Number of cases (%).

**Table 3 jcm-11-00127-t003:** Comparison of the characteristics of the population submitted to pelvic floor rehabilitation on the basis of baseline PAD weight (*ANOVA one-way test*).

	Baseline Pad Weight (g)			*p* Value
	<100 g	101–400 g	>400 g	
Patients, no. (%)	21 (29.2)	25 (34.7)	26 (36.1)	-
Age (years) mean ± SD median	65.5 ± 5.5 65	64.7 ± 65.1 65	66.9 ± 4.2 68	0.1571
Weight (Kg) mean ± SD median	79.5 ± 9.2 83	83.2 ± 9.4 82	81.2 ± 8.9 84	0.4743
BMI mean ± SD median	25.2 ± 1.9 26	26.0 ± 3.1 25	25.5 ± 1.5 25	0.5387
Metabolic Syndrome, no. (%) no mild full	4 (19.1) 15 (71.4) 2 (9.5)	4 (16.0) 18 (72.0) 3 (12.0)	3 (11.5) 18 (69.2) 5 (19.2)	0.1769
Prostate Volume (cc) mean ± SD median	35.1 ± 12.0 34.0	44.8 ± 14.4 45.0	56.9 ± 21.2 56.5	0.0001
Presence of intravesical prostatic lobe, no. (%)	6 (28.6)	4 (16.0)	6 (23.1)	0.9889
Pre-operative total PSA (ng/mL) mean ± SD median	7.3 ± 2.3 7.4	8.6 ± 4.6 7.4	8.1 ± 4.0 7.4	0.6045
Post-operative total PSA (ng/mL) mean ± SD median	0.03 ± 0.02 0.03	0.03 ± 0.01 0.03	0.09 ± 0.21 0.03	0.4361
NS technique at RP, no. (%)	6 (28.6)	8 (32.0)	1 (3.8)	0.0499
Surgical technique at RP, no. (%) LRP RARP	16 (76.2) 5 (23.8)	25 (100.0) 0 (0)	23 (88.5) 3 (11.5)	0.1369
eLND performed at RP, no. (%)	1 (4.8)	7 (28.0)	6 (23.1)	0.3166
Pathological stage (T), no. (%) pT2 pT3 a pT3 b	17 (76.2) 2 (9.5) 2 (9.5)	20 (80.0) 5 (20.0) 0 (0)	16 (61.5) 8 (30.8) 2 (7.7)	0.1648
Positive SM at surgery (R1), no. (%)	4 (19.0)	3 (15.0)	2 (7.7)	0.4684
ISUP grading, no. (%) 1 2 3 4 5	5 (23.8) 10 (47.6) 4 (19.1) 2 (9.5) 0	7 (28.0) 11 (44.0) 4 (16.0) 2 (8.0) 1 (4.0)	7 (26.9) 10 (38.5) 6 (23.1) 3 (11.5) 0	0.6858
Rehabilitation: number of procedures mean ± SD median	8.7 ± 3.9 8.0	12.6 ± 4.7 12.0	15.2 ± 3.8 15.0	<0.0001
Rehabilitation: time length (weeks) mean ± SD median	4.5 ± 1.9 4.0	6.4 ± 2.3 6.0	6.4 ± 2.3 8.0	<0.0001
Pad weight at 2 weeks (g) mean ± SD median	8.5 ± 10.3 5.0	82.3 ± 72.4 70.0	456.8 ± 250.6 410.0	<0.0001
Pad weight at 4 weeks (g) mean ± SD median	7.5 ± 10.1 4.0	49.2 ± 59.3 28.5	272.79 ± 202.64 238.0	<0.0001
Pad weight at 6 weeks (g) mean ± SD median	7.5 ± 9.6 5.0	23.9 ± 28.6 14.0	149.9 ± 130.7 127.0	<0.0001
Pad weight at 12 weeks (g) mean ± SD median	1.6 ± 3.6 0	23.4 ± 44.7 9.0	107.4 ± 112.7 63.5	<0.0001
Pad weight percentage reduction from baseline to 12 weeks, %	92.6	85.0	86.5	
Pad-free cases at 12 weeks, no. (%)	16 (76.2)	10 (4.0)	2 (8.0)	<0.0001

Mean ± SD, median; Number of cases (%); RP = radical prostatectomy; BMI = body mass index; PSA = prostate-specific antigen; LRP = laparoscopic RP; RARP = robot-assisted RP; eLND = extended lymph node dissection; ISUP = International Society of Urological Pathology; SM = surgical margins; NS = nerve-sparing.

**Table 4 jcm-11-00127-t004:** Univariate and multivariate stepwise regression model analyses regarding predictive value of different characteristics in terms of pad test results (pad weight at 12-week follow-up > 10 g) (odds ratio (OR), 95% confidence interval (CI) and *p* value).

	Univariate			Multivariate		
Covariates	OR	95%CI	*p* Value	OR	95%CI	*p* Value
Age (years) - <60 - 61–70 - 71–75	Ref 1.67 1.33	– 0.34–8.17 0.19–9.31	– 0.529 0.772			
Weight (Kg) - <70 - 71–80 - >80	Ref 0.57 0.57	– 0.12–2.75 0.14–2.62	– 0.485 0.425			
BMI - 20–25 - 26–30 - 31–35 - >35	Ref 0.55 0.35	– 0.19–1.53 0.03–4.23	– 0.249 0.412			
Metabolic syndrome - no - mild - full	Ref 1.97 4.08	– 0.51–7.56 0.66–25.38	– 0.324 0.131			
Pre-operative total PSA (ng/mL) - <4.0 - 4.0–10.0 - >10.0	Ref 0.21 0.45	– 0.02–2.05 0.04–5.21	– 0.181 0.523			
Prostate volume - ≤40 cc - >40 cc	Ref 7.84	– 2.71–22.64	– **0.001**	Ref 5.69	– 1.52–21.30	– **0.010**
Presence of endovesical lobe - no - yes	Ref 1.20	– 0.39–3.66	– 0.753			
NS procedure - no - yes	Ref 0.87	– 0.28–2.63	– 0.801			
eLND - no - yes	Ref 1.80	– 0.54–6.03	– 0.340			
ISUP grading - 1 - 2 - 3 - 4 - 5	Ref 0.85 1.91 2.43 3.32	– 0.27–2.70 0.46–7.83 0.39–15.22 0.03–334.92	– 0.787 0.369 0.342 0.611			
pT stage - pT2 - pT3 a - pT3 b	Ref 2.08 1.04	– 0.62–6.90 0.14–7.93	– 0.233 0.971			
Baseline pad weight (g) - <100 g - 101–400 g - 400 g	Ref 9.43 44.00	– 2.28–39.03 8.84–218.99	– **0.002** **<0.0001**	Ref 8.33 35.45	– 1.81–38.17 6.52–192.67	– **0.006** <**0.0001**

OR = odds ratio; CI = confidence interval; BMI = body mass index; PSA = prostate-specific antigen; eLND = extended lymph node dissection; ISUP = International Society of Urological Pathology; NS = nerve sparing. *p* values with bold format indicate significant.

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
