# Peer review of "How to Predict Outcomes from a Biofeedback and Pelvic Floor Muscle Electric Stimulation Program in Patients with Urinary Incontinence after Radical Prostatectomy"

_jcm, 2021, doi:10.3390/jcm11010127_

Round 1

Reviewer 1 Report

The authors reports on the prediction of outcomes of pelvic floor rehab for incontinence after radical prostatectomy. They found that prostate volume was accoaited with increased pad weight after rehabilitation.

This is a well-resented study. The authors may want to revise the beginning of the Results section, however, as it reads more lik a description of RRP outcomes than addressing the core of the study which was analyzing incontinence status. Also, the prostate volume of 40g is arbitrary and it would be interesting to see how other cut offs (or multiple groups) would impact the data. At the minimum, the authors need to justify their cutoff of 40g. Lastly, the authors make it sound that prostate volume is the main contributor to incontinence but the etiology is so multi-factorial that they should address this.

Author Response

Thank you for the positive and useful comments from the Reviewer.

We improved English on minor spelling check.

  1. As requested in the Results section we removed the description related to RRP technique and we maintain only data useful for the actual analysis on UI.
  2. As requested we explained how we decided to stratify our population on the bases on Prostate volume. As explained this is the first article considering preoperative baseline characteristic of a population with post RP UI submitted to rehabilitative treatment. Therefore we had no other previous experience to consider as model. Moreover our population, as underlined in Discussion section, is limited in number and an evaluation of different PV quartiles to find the best stratification was not possible. therefore, as explained in Discussion section page 12 we arbitrarily stratified our population in only two groups ( considering the total number of cases multiple stratification were not considered) and we considered 40 cc as cut-off for the following reasons: a. 40 cc   is  considered in several trials the volume at which the impact of prostate enlargement on bladder outlet   start its negative effect.A significant enlargement of the prostate can be considered to begin at 40 cc b. in this way the number of our cases in each group was well balanced ( 29 and 43 cases respectively). in our opinion this is a good point of stratification so results in terms of pad weight at baseline and response to treatment significantly changed in the two groups
  3. On the contrary the stratification of our population in groups on the basis of baseline pad weight was performed following the indications obtained from our recent meta-analysis on rehabilitative treatment for post RP-UI. In that meta-analysis we found a significant heterogeneity of results among the different groups of treatment . We found a consistently positive association between higher baseline mean pad weight and subsequent improved SMD recovery over the follow-up. A subgroup analysis stratifying for the quartiles of baseline pad weight distribution resulted in a significant reduction in the heterogeneity findings for each single follow-up interval, corroborating the role of initial incontinence variability as a critical predictor among the studies included The best stratification was obtained at the 3 cut point used in the present study so to justify this stratification. Also this point now is underlined in Discussion page...
  4. As requested in discussion page 12. we underline that PV is probably the major factor able to condition results but it is not possible to exclude the multiple influence of other factors and new analysis are recommended

Reviewer 2 Report

The authors present a prospective trial and describe factors that could affect the results of biofeedback and pelvic floor muscle electric stimulation in patients with urinary incontinence that underwent radical prostatectomy due to prostate cancer. The structure is correct, but some questions were raised while studying it.

Patients and Methods:

Conservative treatment for urinary incontinence:

The authors describe that the conservative treatment began 30 days after the catheter removal. Did you do that because you wanted operation trauma relief? In my opinion, you should include the reason.

Statistical analysis:

The parameters studied should be referred as citation.

Results:

The authors refer that they divided the patients in two groups according to the prostate volume and in three groups according to the results of the pad test. Could you please inform us about the criteria you used to divide the patients to groups? That should be referred in the text.

In the Introduction the authors tell us that all diagnostic and therapeutic procedures reflected their routine clinical practice in a Department at high-volume for the management of PC disease. Later, they describe that 34.7% of the patients had a pad test of >400gr 30 days after the removal of the catheter. Unfortunately, this percentage is high enough for LARP and RARP.

Finally, the study is indeed brand new, although it is already known that patients with big prostate glands and middle prostate lobe are more likely to present with worse urinary incontinence. It is promising that departments perform BF and pelvic floor muscle electric stimulation to patients with incontinence after radical prostatectomy. More studies are needed, in order to help patients to get rid of urinary incontinence.

Author Response

We thank the Reviewer for the positive and useful comments.

English spell check has been improved in the manuscript

  1. yes in our experience we always Strat our rehabilitative treatment after 30 days from catheter removal to give time to relief from surgical trauma and also to give the patient to understand the new anatomic condition and appreciate whether a rehabilitative program is needed.. now this point is justify at page 3.
  2. As requested now in Discussion page 12 we explain our choice for stratification on the basis of Prostate volume and baseline pad weight. Prostate volume was chosen arbitrarily considering 40 cc as the point at which enlargement begin its negative effect on bladder outlet in several trials. Baseline pad weight was chosen based on the analysis performed in a previous meta-analysis on rehabilitative treatments for post RP UI in which heterogeneity was reduce stratifying baseline pad weight in the same way.
  3. regarding your observation on the percentage of 34.7% of cases with > 400 cc leakage after 30 days from catheter removal, you must consider that this is they percentage on few cases that need a rehabilitative program when considered to a whole population of cases submitted to the new techniques with a laparoscopic or robotic approach. On approximately 400 cases performed in 2 years, 72  cases (18%) required a rehabilitative program whereas all the other were completely dry after catheter removal  Ot these 72 cases 34% had a sever leakage. These data in our opinion are in line with a department at high volume. This is better underlined at the end of page 2.
  4. At eta end of out manuscript Discussion section we underline that new studied are required 

Round 2

Reviewer 1 Report

Reasonable corrections made.